# Exam stress and the metacognitive strategies of reading in students with dyslexia: The role of motivational mechanisms and educational support

**Kamil Kuracki** [1]*, **Agnieszka Dłużniewska** [2]

**1** Cardinal Stefan Wyszyński University in Warsaw, Warsaw, Poland, **2** Maria Grzegorzewska University in Warsaw, Warsaw, Poland

☯ These authors contributed equally to this work.
* k.kuracki@uksw.edu.pl

## Abstract

Despite numerous studies on dyslexia, there is still a lack of empirical data on the factors determining the functioning of students with dyslexia in the context of written exams. Therefore, the aim of the study is to identify the relationship between sources of stress in an exam situation and the reported use of reading strategies by dyslexic students in terms of motivation and therapeutic interventions in the educational environment. This descriptive-analytical study used data from a larger project focused on children with and without dyslexia. The research sample (n = 640) included girls (n = 280) and boys (n = 360) aged between 14 and 15 years (M = 14.40, SD = 0.55), attending the 7th or 8th grade in Polish mainstream primary schools. Random and intentional sampling was used. All students completed four questionnaires. The results were analyzed using regression analysis in Model templates for PROCESS v4 for SPSS by Hayes. The study showed significant weak and moderate positive correlations between the sources of exam stress and the reading strategies reported by students, ranging from 0.186 to 0.570, as well as significant moderate and strong correlations between reading strategies and experienced educational support, ranging from 0.229 to 0.505, and between reading strategies and motivation to read, ranging from 0.582 to 0.701. The type of stress source significantly influenced the selection of specific reading strategies. Motivation acted as a mediator, while educational support was a moderator in the relationship between exam stress and the reported use of reading strategies. Based on our results, the source of stress may be perceived as factors activating metacognitive mechanisms aimed at selecting appropriate strategies for working with texts. Researchers and teachers should be aware of the need to undertake activities to support students with dyslexia focused on developing the ability to recognize the sources of exam stress and select effective coping strategies.

**Data Availability Statement:** All relevant data are within the paper.

**Funding:** The author(s) received no specific funding for this work.

**Competing interests:** The authors have declared that no competing interests exist.

## Introduction

Dyslexia is a reading disability that is characterized by difficulties in mastering the relationship between the spelling patterns of words and their pronunciations. Dyslexia is a lifelong condition that affects students' academic achievement, educational attainment and socio-emotional wellbeing [1–4]. Aside from the otherwise important academic discussion about the causes, diagnosis, and symptoms of this reading disorder [5–7], dyslexia may influence the cognitive, metacognitive, emotional, motivational and social functioning of individuals, with *stress* also playing an important role [8–12]. Minimizing the negative effects of stress on student functioning can be seen as one of the key challenges for educational and therapeutic practice. Research endeavors [8,13] focusing on the importance of implementing supports in improving reading skills in students with reading difficulties, including dyslexia, suggest a moderating role of academic support in the relationship between exam stress and the metacognitive and motivational mechanisms of reading in dyslexic students. This is particularly important for adolescent students with dyslexia, who developed their literacy skills in the earlier stages of education, and who, upon reaching higher grades of primary school, have to use their often limited literacy skills in producing written assignments and passing written exams as required in the curriculum. The current study addresses this important area of research by investigating the relationship between sources of stress in an exam situation and the reported use of reading strategies by dyslexic students in terms of motivation and therapeutic interventions in the educational environment. These results may potentially complement the area of knowledge about how dyslexic students function in the context of written exams and become the basis for guiding the organization of activities supporting the learning process of dyslexic students.

### Exam stress and dyslexia

Reading and writing can be seen as challenges that trigger negative emotions, especially for dyslexic students. Undoubtedly, one of these negative emotions is exam stress and the related distress of being assessed, especially when performing varied test tasks [14]. Referring to the broader transactional theory of stress [15], this stress may be perceived as a specific kind of relationship between the student and the requirements of the school environment, seen by students as straining their mental resources or threatening their wellbeing. Most children experience this stress, but students with dyslexia may experience it to a greater degree as they have higher levels of perceived generalized anxiety compared to children without dyslexia [16,17]. The sources of exam anxiety may be traced back to, among other things, students' personality-related internal resources. These include a lack of self-confidence and a low level of perceived self-efficacy, which are often secondary symptoms of dyslexia [18]. Furthermore, one of the key symptoms of anxiety is the preoccupation with personal thoughts or emotions, which in an exam situation may result in mind-wandering [19] and, consequently, interfere with both reading comprehension and correct inference [13,20]. Exam anxiety may also be the result of external factors, related both to the unpredictability of the exam situation and to the level of difficulty and the type of task.

Experiencing learning difficulties often leads to anxiety [12,21,22]. Numerous studies show a significant relationship between exam anxiety and the cognitive, emotional and social functioning of students with poor reading skills, including dyslexia, as well as their reading styles and achievements in reading and learning [9,10,23]. Further, studies have demonstrated a link between exam anxiety and self-regulation [24,25]. Interesting data may be found in, for example, Wang et al. [22], who researched a group of 238 students and found a strong association between exam anxiety and reading problems in children with specific learning difficulties. The intimidating feeling of being assessed, alongside the perceived stress, may lead to problems with attentiveness, selectiveness and switching attention, as well as a decreased capacity of

operating memory, all of which are factors directly engaged in the reading process [26,27]. In such a situation, it is crucial to become aware of the sources of anxiety and its consequences for dyslexic students.

## The use of reading strategies and the value of metacognitive knowledge for students with dyslexia

Educational success of students with dyslexia is increased by therapeutic interventions that target both anxiety associated with undertaking school activities and metacognitive strategies that aim to increase the effectiveness of reading [28,29].

Metacognition refers to thinking about one's own thinking [30]. Metacognition contains several aspects, namely, metacognitive knowledge, metacognitive experience and metacognitive and cognitive strategies, which influence each other, illustrating the dynamic nature of metacognition. Metacognition is important in the learning process, such as learning to read and write [31,32]. When reading, students may be oriented to use a variety of strategies that can facilitate their understanding of the text and performance on a variety of school tasks, including exams. If students are focused on general text analysis, they may use global strategies comprising general analysis of the text being read in relation to both the content (metatextual strategies) and its organization. Other strategies are problem-solving strategies, which refer to activities undertaken by students when the text becomes difficult and the reader encounters problems that prevent them from understanding the content. Another group of strategies is auxiliary strategies, which comprise the use of additional aids, such as taking notes, paraphrasing, summarizing and discussing the text [33].

It is also necessary to distinguish between different metatextual strategies [34–36] related to the language system, which focus mainly on solving related problems. These strategies include explaining the meaning of words as a significant predictor for reading [37], syntax-semantic analysis of sentences [38–41] and the strategies for working with the text, which include underlining, rereading some fragments, visualizing the content, summarising, paraphrasing and establishing the goal of reading the text [42–46].

Findings show, first, that the level of metacognition and the resulting ability to monitor reading comprehension is a significant predictor for reading performance [47]; second, that the use of metacognitive reading strategies is an important predictor for reading and learning achievements [48–51]; and third, that the use of instructions aim to develop metacognitive skills in the reading process results in improved reading performance [49]. Research shows that, compared to students who do not display problems with reading, dyslexic students score lower not only in terms of reading skills, but also metacognitive knowledge regarding the text they read and the reported reading strategy [52]. Moreover, they also generally have lower learning efficiency and lower grades. Yet, it appears that students with dyslexia have comparable insight into their own difficulties and are able to adjust their reading comprehension expectations, much like students without any specific reading difficulties. Both groups show similar sensitivity to metacognitive reading experiences [52]. The obtained results may suggest that interventions focused on increasing the level of metacognitive knowledge and reading strategies (global, support, problem-solving strategies) used by students with dyslexia should increase the effectiveness of their working with texts. It remains unclear whether students with dyslexia use text-based strategies under exam stress, and whether the perceived educational support is related to the selection of appropriate strategies.

## Motivation to read in students with dyslexia

Motivational mechanisms form attitudes towards reading and learning, which in turn play an important role in educational achievement [53,54]. As such, their importance for dyslexic

students cannot be overlooked. An appropriate level of motivation of dyslexic students to read and learn with high probability may contribute to the enrichment of metacognitive knowledge and, consequently, to the activation of metacognitive reading strategies in the exam situation, which can improve their ability to read texts.

The motivation towards learning in a broad sense may be influenced by many factors, including self-confidence, persistence, self-esteem and displayed aspirations. The level of difficulty of the exam tasks given to students is also important for activating their motivational mechanisms. In addition to the aforementioned factors, the importance attached to reading and the social context are also emphasized. Although these elements are not always direct stimuli that convince a person to take part in reading activities, they may act as mediators to indirectly influence the motivation of readers [55–57].

In the context of motivating dyslexic students to read, both their conviction about the skills they possess and the availability of social support allow them to believe that they can read successfully and enjoy it [57], thus strengthening their motivation and intensifying reading activities. Shaping and activating the resources of students with reading difficulties, including dyslexia, therefore requires thoughtful, multifaceted educational support [52,58–65] to achieve long-term outcomes. Factors related to the student's awareness of their own motivation may be mediators of the relationship between exam stress and their choice of strategy to read a text. Therefore, cognitive and metacognitive resources, as well as the source of experienced exam stress, should be taken into account when considering reading strategies used by people with dyslexia.

## Educational support for students with dyslexia

Burden and Burdett [66] studied the factors related to dyslexic students' perceptions of their success in learning, and found that properly organized environmental support may positively influence students' motivational convictions about self-efficacy, goal-orientation and sense of control. Such support takes into account elements such as students' attitudes towards the learning process and tasks provided, the way they think about themselves as learners, and their sense of agency in solving problems encountered while learning.

Educational support may be viewed from two perspectives: objective and subjective. The objective dimension relates to both organized, institutional, psychological and pedagogical assistance, as well as informal support from teachers, therapists, parents and friends. The subjective dimension relates to students' perceptions, assessments, and evaluation of activities supporting the learning process [67].

The effectiveness of support interventions for students with reading difficulties, including dyslexia, whether in the form of specific reading strategies [58] or the implementation of multifaceted and multi-element activities aiming to improve the understanding of texts [60], is varied and remains inconclusive. Most reports indicate a link between the use of interventions and an increase in reading comprehension, which is, obviously, promising. For example, Camahalan's [58] quasi-experiment shows the effectiveness of reading program interventions that aim to introduce dyslexic students to reading strategies and improve reading comprehension. Similarly, Stevens et al. [68] demonstrated the effectiveness of interventions oriented on paraphrasing, identifying text structure and discovering the main idea of texts in a small group of students (4–6 persons) with reading difficulties. These two aforementioned studies considered reading comprehension and the use of specific strategies; therefore, a multi-factor intervention could have cumulative positive results. However, Fogarty et al. [60] had inconsistent observations, finding no statistically significant differences in improved reading comprehension between an experimental group of 411 students and the control group (who continued

usual reading practice) of 488 students. Because of the multifaceted and multi-factor nature of the program, it is possible that the nonsignificant results are due to insufficient intervention time (12 weeks) and not enough meetings (36 lessons, 50 minutes each).

Both theoretical backgrounds and empirical sources give reasons for undertaking research examining the relationship between educational support, motivation and activation of metacognitive mechanisms and choosing an appropriate reading strategy and exam stress.

## The current study

The research conducted to date has mainly focused on the effectiveness of support to increase motivation and results in learning and school functioning among students with dyslexia. However, there are no studies investigating a) the mediating role of motivation as an internal resource of dyslexic students, nor b) the moderating role of educational supports on the relationship between the source of exam stress and the activation of metacognitive reading strategies. Our research aims to fill these gaps and address the following research questions:

1. What sources of stress, reading strategies, motivation and experienced educational support are reported by students with dyslexia?

2. Are there associations between stress factors in the exam situation and the use of reading strategies (global, supportive and problem-solving strategies in reading) as reported by dyslexic students, their motivation to engage in the reading activity and their perceived educational support?

3. How do various sources of exam stress relate to dyslexic students' choices of specific reading strategies?

4. Does the motivation to read and perceived educational support among students with dyslexia mediate the relationship between the sources of exam stress and reported use of metacognitive reading strategies?

Taking into account the negative impact of stress on school achievements confirmed in research [13,20], we hypothesize that the source of stress may determine the choice of specific reading strategies in an exam situation. Considering that school achievements are determined both by internal (e.g., motivation) and external (e.g., educational support) factors [57,58,69,70], we hypothesize that motivation will be a mediator, and educational support will be a moderator, of the relationship between the independent variable (stress) and the dependent variable (choice of reading strategy) in the exam situation.

## Methods

The presented descriptive-analytical research is part of a wider project (implemented in the years 2015–2021) examining the reading practice, participation in culture and selected psychological resources of students with and without dyslexia in adolescence, including metacognitive awareness and motivational mechanisms of reading during exam stress. The cohort comprised N = 1820 students from 73 classes of 38 Polish primary schools, including students with dyslexia (N = 640) and without dyslexia (N = 1180). The study used a random selection of schools and classes. The selection of students from each class was intentional. Students diagnosed with dyslexia were qualified for the study. After the schools were selected, the consent of the school principals was requested, and then the consent of the students' parents and students. Parents and students were informed about the purpose and conditions of the study and that they may, at any time, without giving any reason, refuse to participate in the study. The study received ethics permission from Research Ethics Committee of The Maria Grzegorzewska

University (No 106-2014/2015). All parents/guardians of students from the study group gave their written consent for their child's participation in the study beforehand. They were informed that the test was one-off, anonymous and conducted in a safe and non-invasive way at school (e.g., during a class). They were also informed that students may ask for a break or withdraw from the study at any time, without giving a reason, and that doing so will not negatively impact students.

## Participants

The current study comprised 640 students (280 girls, 360 boys), between 14 and 15 years of age (M = 14.40, SD = 0.55), attending the 7th or 8th grade (mainstream primary school). In the Polish education system, these are the last two years of primary school, at the end of which students sit an exam; the results of this exam determine their eligibility for further education.

A power calculation indicates that the size of the research sample is sufficient. Considering that nearly 10% of the 5 million Polish students have dyslexia [71], assuming a 95% confidence interval for the study and a maximum error of 5%, using the Taro Yamane formula, the minimum sample size should be N = 400.

Participants were recruited for the study between June 2016 and December 2021. All students were diagnosed with dyslexia in accordance with the ICD-10 [72] criteria by teams of specialists. In Poland, a diagnosis of dyslexia is carried out in Psychological and Pedagogical Counseling Centers, in which specialists such as Psychologists, Educators, or Speech Therapists, using standardized tools, determine the cause of difficulties in reading and writing. These tools examine the level of perceptual-motor functions (visual and auditory analysis), rapid naming, memory, attention, literacy (including decoding and reading comprehension) and IQ of students. As a result, students who meet the dyslexia diagnostic criteria receive a document confirming that the cause of their reading and writing difficulties is dyslexia. This document is included in school records. In the present study, only students whose dyslexia diagnosis was confirmed by the School Head, based on school records, were eligible to participate. The authors of the study obtained only information about the diagnosis of dyslexia, did not review any other medical records of the child and did not have access to information that could identify individual participants during or after data collection.

## Measures

Four psychometric tools were used in the study. Each of them showed high Cronbach's alpha internal consistency coefficient values for entire questionnaires and their individual subscales, both in the original validation sample and in the analyses within this study. All original versions of research tools used in the study were linguistically adapted and validated by authors (two-sided translation by 3 independent experts) with the consent of the authors.

## Sources of anxiety

The Students' Perceived Sources of Test Anxiety questionnaire [73] was used to identify the sources of anxiety experienced by the students. The tool uses a 5-point Likert scale (1 = "not at all", 5 = "very strongly"), within which the participants rate the extent to which each item reflects their emotions and thoughts before taking an exam. The tool consists of 22 items placed within seven subscales: test uncertainty, test difficulty, instrumentality of test results, low self-efficacy, temporary deficit, anxiety proneness, and situational uncertainty. As outlined in the manual, the results may be interpreted in relation to individual subscales or in relation to the following three master factors: Factor I, "perceptions of the test" (uncertainty, test difficulty and instrumentality of test results) showed a Cronbach's alpha value in this research

sample of 0.76; Factor II "perceptions of the self" (low self-efficacy, temporary deficit, and anxiety proneness), with its reliability reaching 0.88; Factor III "perception of the situation" (situational uncertainty), with its reliability in the test sample equal to 0.72.

## Reading strategies

The self-report tool known as the Metacognitive Awareness of Reading Strategies Inventory (MARSI) [33] was used to measure the students' reported use of reading strategies. The tool uses a 5-point Likert scale (1 = "I never or almost never do it"; 5 = "I do it always or almost always"). The validation of the tool, which originally contained 60 items, was carried out on a group of 825 pupils (52.8% female) in grades 6 to 12. Cronbach's alpha coefficient for the entire inventory was 0.93 and ranged from 0.89 to 0.93 for individual subscales. The final, revised version contained 30 items, and the reliability of the tool for individual subscales was estimated on a sample of 443 students reached for each subscale: Global Reading Strategies = 0.92, Problem-Solving Strategies = 0.79 and Support Reading Strategies = 0.87. Reliability for all items equals 0.89. In the present sample, the reliability for individual subscales was as follows: Global Reading Strategies = 0.87, Problem-Solving Strategies = 0.84 and Support Reading Strategies = 0.86.

## Motivation for reading

The Motivations for Reading Questionnaire [74] was used to measure the students' reported motivation to read. The tool uses a 4-point Likert scale (1 = "very different from me", 2 = "a little different from me", 3 = "a little like me", and 4 = "a lot like me"). The first version of the tool contained 82 items; the revised version had 54 items [75]. The tool consists of 11 subscales, with the Cronbach's alpha reliability coefficient ranging from 0.43 to 0.80. The current study used a tool after the second revision, which, on the basis of factor analysis, identified 37 items and assigned eight subscales: Social, Grades–Compliance, Curiosity, Competition, Involvement, Reading work avoidance, Efficacy and Recognition [76]. The global score, which is the sum of individual factors excluding the Reading work avoidance subscale, may also be included in the analysis. For this research sample, the Cronbach's alpha reliability coefficient reached 0.93, while the values for individual subscales ranged from 0.62 to 0.87. Performance on all subscales are reported. However, the global scale was used for analysis.

## Educational supports

The Educational Support Questionnaire (ESQ) [67] assesses the students' perception of support provided by teachers, parents, peers, and specialists from psychological and pedagogical counseling centers in various educational situations using a 5-point Likert scale (1 = "completely false", 5 = "completely true"). The tool consists of a total of 30 items. The main part of the questionnaire contains 24 items within three subscales: educational support from teachers, parents, and classmates. Each holds eight items. The additional part of the ESQ contains six statements concerning educational support provided by specialists from a psychological and pedagogical counseling center or a school as part of additional specialist classes, e.g., remedial classes. The validation study of the questionnaire showed that the split-half reliability coefficient values ranged from 0.79 to 0.91. The Cronbach's alpha coefficients for individual subscales ranged from 0.78 to 0.92. The reliability of the tool for this research sample for the global educational support scale was 0.92, for the subscales: educational support from teachers = 0.88, educational support from parents = 0.85, and educational support from classmates = 0.91.

## Results

### Data analysis

The results were analyzed using regression analysis in Model templates for PROCESS v4 for SPSS by Hayes. The analysis established relationships between stressors in the exam context and particular reading strategies (global, support and problem-solving reading strategies), as well as students' motivations for engaging in reading and their perceived support. Statistically significant correlations between these variables became the basis for regression analyses that aimed to identify the factors determining the dyslexic students' choice of reading strategies and the mediating factors in the relationship between the sources of exam stress and the use of specific reading strategies. The main assumptions of regression analysis regarding, among others, the quantitative nature of the variables, normality of distribution, the ratio of the number of observations to the analyzed variables and the VIF coefficient values were met.

### The sources of stress, reading strategies, motivation and experienced educational support in students with dyslexia

To answer the first research question, descriptive statistical analyses were performed. As shown in Table 1, the most frequently reported sources of exam stress related to Perception of the Self (M = 2.973, SD = 0.88) and Perception of the Test (M = 2.918, SD = 0.72). Perception of the Situation was seen as a source of exam stress less frequently (M = 2.438, SD = 1.06). In the case of metacognitive reading strategies, participants reported using problem-solving strategies the most frequently (M = 3.457, SD = 0.84), and support reading strategies the least frequently (M = 2.743, SD = 0.95). Considering the average range of subscales of the Global Motivation variable, cognitive curiosity (M = 2.809, SD = 0.71) and involvement (M = 2.630, SD = 0.89) were the most frequently indicated sources of motivation, with social factors being the least important (M = 2.094, SD = 0.59). In terms of educational support, students were least likely to feel supported by their peers (M = 23.450, SD = 8.42), while they were most likely to feel supported by their parents (M = 28.840, SD = 6.86).

### Relations between stress factors in the exam environment, using reading strategies, motivation, and experienced educational support in students with dyslexia

To answer the second research question, Pearson's correlational analyses were performed to determine the relationship between exam-related stress factors, reading strategies (Global Reading Strategies, Support Reading Strategies, Problem-Solving Strategies), motivation for taking up a reading activity (Global Motivation) and perceived educational support from various sources (Global Education Support, Educational Support from Teachers, Educational Support from Parents, Educational Support from Peers) (Table 2).

The analyses showed significant weak-to-moderate positive correlations between individual sources of exam stress and Global Motivation, ranging from 0.175 to 0.317. The analyses also revealed moderate-to-strong positive correlations between Global Motivation and all reading strategies used by students, ranging from 0.582 to 0.701. This motivation was also significantly positively correlated with individual sources of educational support, particularly with the Global Education Support (r = 0.497, p<0.01) and the Educational Support from Teachers (r = 0.407, p<0.01). Moreover, the results showed moderate positive correlations between Perception of the Test and specific reading strategies, ranging from 0.408 to 0.570, as well as between Perception of the Self and the reported use of Support Reading Strategies (r = 0.368, p<0.01) and Problem-Solving Strategies (r = 0.362, p<0.01). The Perception of the Situation

**Table 1. Mean results for individual variables (N = 640).**

| Variable | M | SD | Min | Max |
|---|---|---|---|---|
| *Students' perceived sources of test anxiety* | | | | |
| Perception of the Test | 2.918 | 0.72 | 1.00 | 4.67 |
| Test uncertainty | 2.703 | 0.79 | 1.00 | 4.25 |
| Test difficulty | 3.148 | 1.05 | 1.00 | 5.00 |
| Instrumentality of test results | 3.052 | 1.02 | 1.00 | 5.00 |
| Perception of the Self | 2.973 | 0.88 | 1.00 | 4.82 |
| Low self-efficacy | 2.980 | 0.85 | 1.00 | 5.00 |
| Temporary deficit | 2.766 | 1.49 | 1.00 | 5.00 |
| Anxiety proneness | 3.003 | 1.00 | 1.00 | 5.00 |
| Perception of the Situation | 2.438 | 1.06 | 1.00 | 5.00 |
| Situational uncertainty | 2.438 | 1.06 | 1.00 | 5.00 |
| *Motivations of Reading Questionnaire* | | | | |
| Global Motivation | 2.432 | 0.57 | 1.00 | 3.68 |
| Social | 2.094 | 0.59 | 1.00 | 3.50 |
| Grades – Compliance | 2.459 | 0.71 | 1.00 | 4.00 |
| Curiosity | 2.809 | 0.71 | 1.00 | 4.00 |
| Competition | 2.395 | 0.85 | 1.00 | 4.00 |
| Involvement | 2.630 | 0.89 | 1.00 | 4.00 |
| Reading work avoidance | 2.589 | 0.75 | 1.00 | 4.00 |
| Efficacy | 2.272 | 0.69 | 1.00 | 3.60 |
| Recognition | 2.516 | 0.93 | 1.00 | 4.00 |
| *MARSI* | | | | |
| Global Reading Strategies | 3.077 | 0.81 | 1.00 | 4.38 |
| Support Reading Strategies | 2.743 | 0.95 | 1.00 | 4.67 |
| Problem-Solving Strategies | 3.457 | 0.84 | 1.00 | 4.75 |
| *The Educational Support Questionnaire* | | | | |
| Global Educational Support | 96.190 | 20.39 | 30.00 | 142.00 |
| Educational Support from Teachers | 24.160 | 6.95 | 8.00 | 39.00 |
| Educational Support from Parents | 28.840 | 6.86 | 8.00 | 39.00 |
| Educational Support from Peers | 23.450 | 8.42 | 8.00 | 40.00 |

M = arithmetic mean, SD = standard deviation.

as a source of exam stress was positively correlated with all metacognitive reading strategies at a low-to-moderate level, ranging from 0.186 to 0.312.

The analyses also showed weak positive correlations between Perception of the Test and Global Educational Support (r = 0.252, p<0.01) as well as between Perception of the Self and Global Educational Support (r = 0.127, p<0.01). However, no statistically significant relationships were found between the Perception of the Situation and the perceived level of

**Table 2. Correlations (r = Pearson) the examined variables (N = 640).**

| | 1 | 2 | 3 | 4 | 5 | 6 | 7 | 8 | 9 | 10 |
|---|---|---|---|---|---|---|---|---|---|---|
| 1. Perception of the Test | | | | | | | | | | |
| 2. Perception of the Self | 0.747** | | | | | | | | | |
| 3. Perception of the Situation | 0.473** | 0.358** | | | | | | | | |
| 4. Global Motivation | 0.317** | 0.306** | 0.175** | | | | | | | |
| 5. Global Reading Strategies | 0.570** | 0.421 | 0.267** | 0.701** | | | | | | |
| 6. Support Reading Strategies | 0.436** | 0.368** | 0.312** | 0.582** | 0.582** | | | | | |
| 7. Problem-Solving Strategies | 0.408** | 0.362** | 0.186** | 0.611** | 0.611** | 0.723** | | | | |
| 8. Global Education Support | 0.253** | 0.127** | 0.031 | 0.497** | 0.497** | 0.505** | 0.503** | | | |
| 9. Educational Support from Teachers | 0.092 | -0.098* | -0.049 | 0.407** | 0.407** | 0.377** | 0.443** | 0.798** | | |
| 10. Educational Support from Parents | 0.132** | 0.109** | -0.068 | 0.295** | 0.295** | 0.266** | 0.261** | 0.675** | 0.378** | |
| 11. Educational Support from Peers | 0.222** | 0.112** | 0.134** | 0.280** | 0.280** | 0.365** | 0.229** | 0.738** | 0.385** | 0.310** |

** correlations $p < 0.01$

* correlation at $p < 0.05$.

educational support. It should be highlighted that the Perception of the Situation as a source of exam stress displayed significant positive correlations, at the weak level, only with educational support from peers ($r = 0.134$, $p<0.01$). In contrast, there were weak positive correlations between test perception and three of the sources of educational support, ranging from 0.132 to 0.253. Surprisingly, Perception of the Self correlated positively (weak correlation) with educational support received from family ($r = 0.109$, $p<0.01$) and from peers ($r = 0.112$, $p<0.01$), but negatively (weak correlation) with support received from teachers ($r = -0.098$, $p<0.05$).

The analysis allows for an observation that all sources of educational support, at a weak-to-moderate level, are significantly and positively related to students' reported reading strategies, ranging from 0.229 to 0.505. However, it should be noted that particularly high correlations were discovered between reading strategies and the support received from teachers, ranging from 0.377 to 0.443. Furthermore, the study revealed moderate-to-high positive correlations between particular metacognitive reading strategies, individual sources of exam stress and specific sources of educational support.

To answer the third and fourth research questions, a regression analysis was performed based on the assumption that motivation may act as a mediator between the independent variable (exam-related stress) and the dependent variable (choice of reading strategy), while educational support may be a moderator of this aforementioned relationship. The analysis found that three models were well-fitted to the data.

For two of these models, the independent variables were perceiving oneself as a source of exam stress (Perception of the Self) and Perception of the Situation, while the dependent variable was global reading strategies. In both cases, the facilitator role belonged to Global Motivation, and the facilitator role belonged to educational support. The first model explained 58% of the variance of the dependent variable (Table 3), with the second model accounting for almost 55% of the variance of the dependent variable (Table 4).

**Table 3. Statistics describing the percentage of variance accounted for by the predictors of the dependent variable $R^2$.**

| | | | | | | | Model 1 –summary | | | | |
|---|---|---|---|---|---|---|---|---|---|---|---|
| R | $R^2$ | MSE | F | df1 | df2 | p | Change statistics | | | | |
| | | | | | | | $R^2$ | F | df1 | df2 | p |
| 0.762 | 0.581 | 46. 887 | 219.926 | 4 | 635 | < 0.001 | 0.012 | 18.649 | 1 | 635 | < 0.001 |

**Table 4. Statistics describing the percentage of variance accounted for by the predictors of the dependent variable R $^2$.**

| | | | | | | | Change statistics | | | | |
|---|---|---|---|---|---|---|---|---|---|---|---|
| R | R$^2$ | MSE | F | df1 | df2 | p | R$^2$ | F | df1 | df2 | p |
| 0.741 | 0.549 | 50. 412 | 193.448 | 4 | 635 | < 0.001 | 0.006 | 7.753 | 1 | 635 | 0.006 |

*Model 2 –summary*

In model 1, significant predictors of the dependent variable are Perception of the Self as a source of stress related to the exam situation (B = 0.766; p < 0.001), Global Motivation (B = 0.270; p < 0.001), educational support (B = 0.273; p < 0.001) and the interaction variable of perceiving oneself (Perception of the Self) and educational support (B = -0.005; p < 0.001).

In model 2, significant predictors of the dependent variable are situational uncertainty (Perception of the Situation) as a source of exam stress (B = 2.553; p < 0.001), Global Motivation (B = 0.270; p < 0.001), educational support (B = 0.189; p < 0.001) and the interaction variable of Perception of the Situation and educational support (B = -0.018; p = 0.006).

Tables 5 and 6 present the values of the conditional effects for the predictors' Perception of the Self as a source of stress related to the exam situation and situational uncertainty (Perception of the Situation) depending on the moderator's value (i.e., educational support).

Analyzing the values of the different ranges of educational support indicates that this variable significantly conditions the relationship between both sources of exam stress (i.e., Perception of the Self and Perception of the Situation) and the reported use of global reading strategies in all ranges of students' perceived support. Thus, in the case of dyslexic students and, regardless of the level of perceived support (high, medium, low), the reported use of global reading strategies increases with the rise in the level of both perceiving oneself as a source of stress (Perception of the Self) and situational uncertainty (Perception of the Situation). This indicates that any amount of educational support increases the effectiveness of coping with exam anxiety by means of finding ways to cope with tests, in this case, choosing global reading strategies. It is worth noting that in both cases, the increase in the level of support experienced by students is associated with a drop in the value of the conditional moderator's direct effect on the relationship between the variables Perception of the Self and the reported use of global reading strategies, as well as Perception of the Situation as a source of exam stress and the reported use of global reading strategies.

The standardized values of the indirect influence of motivation on the relationship between Perception of the Self as a source of exam stress and reported use of global reading strategies and the relationship between Perception of the Situation as a source of exam stress and reported use of global reading strategies as obtained by bootstrapping for 5000 repetitions reach the value of 0.15 respectively, with the confidence interval of [0.101; 0.205] and 0.10 for one and the confidence interval of [0.054; 0.139] for the other.

In the third analyzed model, the independent variable was the Perception of the Test, the dependent variable was the support reading strategies, and the mediator, as in the two previous

**Table 5. Estimation of conditional effects of the predictor for moderator values.**

| Educational support | Effect | Standard error | t | p | Confidence interval | |
|---|---|---|---|---|---|---|
| | | | | | Lower limit | Upper limit |
| 78.000 | 0.346 | 0.036 | 9.570 | < 0.001 | 0.275 | 0.417 |
| 99.000 | 0.233 | 0.030 | 7.762 | < 0.001 | 0.174 | 0.291 |
| 114.000 | 0.152 | 0.038 | 3.988 | < 0.001 | 0.077 | 0.226 |

**Table 6. Estimation of conditional effects of the predictor for moderator values.**

| Educational support | Effect | Standard error | t | p | Confidence interval | |
|---|---|---|---|---|---|---|
| | | | | | Lower limit | Upper limit |
| 78.000 | 1.123 | 0.178 | 6.314 | < 0.001 | 0.774 | 1.472 |
| 99.000 | 0.738 | 0.137 | 5.372 | < 0.001 | 0.468 | 1.007 |
| 114.000 | 0.463 | 0.182 | 2.541 | 0.011 | 0.105 | 0.820 |

models, was Global Motivation, while the moderator was educational support (Table 7). The model explains approximately 48% of the variance of the dependent variable.

In the third model, significant predictors of the dependent variable are Perception of the Test (B = -0.480; p = 0.001), Global Motivation (B = 0.195; p < 0.001), educational support (B = -0.126; 0.003) and the interaction variable of Perception of the Test and educational support (B = 0.010; p < 0.001). Table 8 presents the values of the conditional effects for the predictor Perception of the Test depending on the value of the moderator, namely educational support.

The analysis of the values of individual ranges of students' perceived educational support indicates that this variable significantly conditions the relationship between Perception of the Test and the reported use of support reading strategies in all moderator ranges. As dyslexic students' perception of the test as a source of stress increases, the reported use of support reading strategies also increases. The moderating role of educational support should be interpreted slightly differently than in the two models discussed beforehand. As the level of support perceived by students increases, the value of the moderator's direct conditional effect on the relationship between the Perception of the Test as a source of stress and the reported use of support reading strategies increases.

The value of the standardized indirect effect obtained using the bootstrapping method for 5,000 repetitions proves that the value of the indirect role of motivation in the relationship between the Perception of the Test and the reported use of support reading strategies as determined by the students' perception of educational support reaches approximately 0.14, with the confidence interval of [0.096; 0.184].

## Discussion

The current study attempted to identify the relationships between exam-related stressors and dyslexic students' reported use of metacognitive reading strategies, motivation to engage in reading activities and perceived educational support. Another goal was to determine whether reading motivation and educational support perceived by dyslexic students mediated the relationship between sources of exam stress and the reported use of global support and problem-solving reading strategies.

The multi-stage quantitative analysis of the results found many statistically significant correlations between the studied variables. Each potential source of exam stress related to the perception of the test, oneself and the situation was significantly related to the global motivation to read among students with dyslexia. The increase in global motivation to read as a result of experiencing stress during the exam, may be explained by theories of motivation, according to

**Table 7. Statistics describing the percentage of variance accounted for by the predictors of the dependent variable R $^2$.**

| R | $R^2$ | MSE | F | df1 | df2 | p | Change statistics | | | | |
|---|---|---|---|---|---|---|---|---|---|---|---|
| | | | | | | | $R^2$ | F | df1 | df2 | p |
| 0.696 | 0.484 | 38. 310 | 148.984 | 4 | 635 | < 0.001 | 0.027 | 33.750 | 1 | 635 | < 0.001 |

**Table 8. Estimation of conditional effects of the predictor for moderator values.**

| Educational support | Effect | Standard error | t | p | Confidence interval | |
|---|---|---|---|---|---|---|
| | | | | | Lower limit | Upper limit |
| 78.000 | 0.210 | 0.045 | 4.630 | < 0.001 | 0.121 | 0.299 |
| 99.000 | 0.396 | 0.042 | 9.548 | < 0.001 | 0.315 | 0.477 |
| 114.000 | 0.529 | 0.052 | 10.117 | < 0.001 | 0.426 | 0.631 |

which the optimal level of stimulation often facilitates the mobilization to meet one's own requirements and those of the environment [77]. Furthermore, the link between perceiving oneself as a source of stress and global motivation to read appears to relate, to some extent, to research findings suggesting that students with dyslexia may believe that those who read well are seen as more intelligent [78]. Stress resulting from feelings of having inadequate competencies may, in fact, motivate students with dyslexia to undertake activities that stimulate the development of reading skills. Some studies emphasize that the importance assigned to reading and the social context can indirectly influence readers' motivation [55–57]. However, the results of the present study show that students with dyslexia most often indicated cognitive curiosity and engagement as the primary sources of motivation, with the social factor being less important; therefore, their motivation to read was primarily internal.

The analyses also allowed for an observation that, as global reading motivation increases, dyslexic students are more likely to report the use of certain metacognitive reading strategies in the reading process. Similar relationships were discovered between the perception of the test as a source of exam stress and specific reading strategies, as well as between self-perception and supported reading strategies and problem-solving reading strategies. Therefore, the motivation to read seems to be one of the factors that, to a certain extent, may increase the cognitive awareness of students and, consequently, indirectly contributes to their taking up activities conducive to effective learning. Such activities include the selection of appropriate strategies when analyzing the test and seeing their own skills as potential sources of failure. However, according to the correlation analyses, motivation to read was significantly related to specific sources of educational support, especially that from teachers. On the one hand, this suggests that educational support from many sources affects the volition of students with difficulties in learning. On the other hand, it suggests an important role of teachers, who should include supporting dyslexic students as an integral part of the programmed activities aimed at shaping their motivational resources. Such an assumption corresponds with the idea that students' beliefs in the availability of support enable them to believe in their own reading success and strengthen their motivation to read and engage in reading activities [57].

There were weak positive correlations between the perception of the test and the perception of the self as sources of exam stress and global educational support. These findings indicate that students with dyslexia, along with the increased stress related to the perception of the test structure and their own competences, rate the educational support that they received as much higher. This may be explained by the idea that, when faced with an exam situation, students with dyslexia begin to appreciate the importance of specific techniques for working with the text and with their emotions, as they have been taught both at home and at school. This is also confirmed by the positive relationship between students' perception of the test as a source of exam stress and their perception of educational support from teachers, parents, and peers. However, these results conflict with Burden and Burdett [66], who showed that adequate environmental support may have a positive effect on the beliefs of dyslexic students concerning, among other factors, their self-efficacy. In this case, it would be reasonable to expect negative correlations between the assessment of social support and the perception of oneself as a source

of stress. However, the analyses did not find any relationships between the perception of the situation and the perceived level of educational support, which may stem from students' beliefs that they have no real influence on unpredictable factors, as no one is able to prepare for the unexpected. It is worth noting that classmates may be an exception to this rule because they experience the same unexpected situations during exams and may be a significant source of support, especially emotional support.

The negative correlations between perceiving oneself as a source of exam stress and the support received from teachers, alongside the positive correlations between self-perception and support from family and peers, observed in this study are interesting. They suggest, on the one hand, that with lower perceived support from teachers, the stress experienced by students with dyslexia in an exam situation is more likely to stem from their perception of their own skills as insufficient to meet the requirements of the exam. On the other hand, they suggest that, with greater perceived educational support from parents and peers, the stress related to self-perception increases. As discussed earlier, exam-induced stress may lead to mind-wandering, which interferes with effective work with texts [19]. The former scenario suggests that teachers should direct their educational support towards developing adequate self-esteem and metacognitive awareness in their students. The latter, however, suggests that the family and peer environment serve as important reference points for students with learning disabilities, and that it is important for family and peers not to let down those closest to them because they can play a significant role in developing students' learning skills [69,70,79,80] and therefore educational success.

The results of the correlation analyses laid the foundations for a regression analysis, which identified the predictors of triggering metacognitive mechanisms in students with dyslexia, the choice of specific reading strategies, and thus the effectiveness of learning. The source of exam stress determined the choice of specific reading strategies, which confirmed our first hypotheses. This is in line with Furnes and Norman [52], who demonstrated that students with dyslexia had a level of sensitivity to metacognitive reading experiences comparable to students without reading difficulties. This suggests that they are able to correctly assess the exam situation related to the need to solve the test and choose the appropriate strategies for working with a text.

Our study also found that, in a situation where dyslexic students see the source of exam stress in themselves or in situational uncertainty, they report the use of global strategies to increase reading efficiency. These global strategies include activities such as familiarising themselves with text structure, determining the goal for reading and using context or seeking support in typographic guidelines (e.g., tables, diagrams and illustrations). Both motivation and students' perceived educational support are involved in the relationship between the source of stress and the choice of strategies for working with the text. It may be concluded that the feeling of educational support, and activation of motivational mechanisms, contribute to a more task-oriented attitude, involving the search for specific strategies, which can minimize the impact of stress on test-taking performance in an exam situation [81].

When dyslexic students perceive the test as a source of stress, they report using supported reading strategies (i.e., those that focus on the text), which include taking notes, underlining selected passages, summarising, using dictionaries, rereading the text or talking to the teacher (therapist) to check the level of text comprehension. When the test is a source of stress for the students, it can change the role of educational support and motivation in finding and activating metacognitive mechanisms and, therefore, choosing the best reading strategy. Contrary to the previous two instances, in this case, a higher level of support experienced by students can determine the influence of exam stress on the use of supported reading strategies. It is possible to explain these differences in terms of the effectiveness of educational interventions, which

are more likely to be beneficial when the source of stress is the student or the exam situation rather than when it is the test itself. Alternatively, it may stem from better assimilation of strategies for coping with stress (from themselves or the situation) among students partaking in therapy, and such mechanisms were not internalized for cases when the source of stress is the test. Therefore, the above results confirm both hypotheses.

## Conclusions and future directions

We found that the type of stress source determines the selection of specific reading strategies and that motivation is a mediator, and educational support is a moderator, of the relationship between exam stress and the reported use of reading strategies. Therefore, the source of stress should be perceived as a factor activating metacognitive mechanisms aimed at selecting appropriate strategies for working with texts.

Findings show the need for students with dyslexia to undertake activities within the framework of educational support that focus on developing their ability to recognize sources of exam stress and select effective coping strategies. Teaching students with dyslexia to recognize sources of stress and implement proper strategies for working with text should be permanent (not ad-hoc), multifaceted aspects of intervention programs. One such example could entail altering exam protocols to allow such students to use their specific strategies. This seems particularly relevant when the source of stress is the test and its organizational and linguistic structure. Creating conditions for taking notes, summarising, using dictionaries, discussing the content with the teacher or extending the duration of the exam to allow for effective use of strategies for working with the text could enable dyslexic students to achieve higher learning outcomes.

Considering that students with dyslexia perceive themselves as the main source of exam stress, system-based actions should also be undertaken to help them develop an appropriate level of self-esteem and self-effectiveness, activate motivational mechanisms and insight into their own cognitive and psychosocial resources. These interventions could be prioritized to guarantee success in learning.

Identifying the functional determinants of dyslexic students in a stressful exam situation requires a broader examination of students' resources. These can be internal, related to self-perception (including their sense of competence, values and agency), or external, related to the organization of the educational and therapeutic environment and adjustment of didactic resources, including skill tests, to match the diverse abilities of students with dyslexia.

## Limitations

The multifaceted nature and complexity of the research issues addressed in the current project make it impossible to avoid certain limitations. Firstly, only declarative uses of metacognitive strategies were explored. Conducting an experimental study that would monitor the difference between the degree of declarative and actual use of the strategies would allow for more objective conclusions to be drawn.

Secondly, the level of perceived educational support is a function of students' subjective impressions. The project did not control the way support activities were organized and conducted. It can only be assumed that institutional support (teachers, therapists) was organized in accordance with the guidelines contained in the legal regulations on psychological and pedagogical assistance currently in force in Poland. As such, it would have been consistent with the standards of conduct recommended by leading organizations, associations and research centers supporting people with dyslexia all over the world. However, it is difficult to state exactly what support the students received from their families and friends.

Thirdly, the models could be more detailed and take into account specific types of support and particular motivational mechanisms, which would help to direct therapy towards individual areas of students' functioning and environmental organization.

Finally, due to the large number of schools and classes participating in the study, these analyses do not take into account variables such as classroom interactions and school climate, which may impact the results obtained and which would be worth considering in future research.

## Acknowledgments

We are grateful to Prof. Marleen Westerveld (Griffith University, Australia), who reviewed the initial draft of the manuscript.

## Author Contributions

**Conceptualization:** Kamil Kuracki, Agnieszka Dłużniewska.

**Data curation:** Kamil Kuracki, Agnieszka Dłużniewska.

**Formal analysis:** Kamil Kuracki, Agnieszka Dłużniewska.

**Investigation:** Kamil Kuracki, Agnieszka Dłużniewska.

**Methodology:** Kamil Kuracki, Agnieszka Dłużniewska.

**Project administration:** Kamil Kuracki, Agnieszka Dłużniewska.

**Resources:** Kamil Kuracki, Agnieszka Dłużniewska.

**Software:** Kamil Kuracki, Agnieszka Dłużniewska.

**Supervision:** Kamil Kuracki, Agnieszka Dłużniewska.

**Validation:** Kamil Kuracki, Agnieszka Dłużniewska.

**Visualization:** Kamil Kuracki, Agnieszka Dłużniewska.

**Writing – original draft:** Kamil Kuracki, Agnieszka Dłużniewska.

**Writing – review & editing:** Kamil Kuracki, Agnieszka Dłużniewska.

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
