## [Decision Letter · Decision Letter 0]

21 Jun 2023

PONE-D-23-09745Exam stress and the metacognitive strategies of reading in students with dyslexia: the role of motivational mechanisms and educational supportPLOS ONE

Dear Dr. Kuracki,

Thank you for submitting your manuscript to PLOS ONE. After careful consideration, we feel that it has merit but does not fully meet PLOS ONE’s publication criteria as it currently stands. Therefore, we invite you to submit a revised version of the manuscript that addresses the points raised during the review process.

We look forward to receiving your revised manuscript.

Kind regards,

Yuh-Yuh Li, Ph.D.

Academic Editor

PLOS ONE

Journal Requirements:

Reviewers' comments:

Reviewer's Responses to Questions

**Comments to the Author**

1. Is the manuscript technically sound, and do the data support the conclusions?

Reviewer #1: Yes

Reviewer #2: Yes

2. Has the statistical analysis been performed appropriately and rigorously? 

Reviewer #1: Yes

Reviewer #2: Yes

3. Have the authors made all data underlying the findings in their manuscript fully available?

Reviewer #1: Yes

Reviewer #2: No

4. Is the manuscript presented in an intelligible fashion and written in standard English?

Reviewer #1: Yes

Reviewer #2: No

5. Review Comments to the Author

Reviewer #1: This is an excellent study. Valid methods were used in the identification of dyslexia.The authors clearly stated the gap identified in the literature, the need for this study and its implication on the education of students with dyslexia both in Poland and the world. However, the statement on the justification for this study is not strategically positioned in the manuscript to draw readers attention, arouse their curiosity and sustain their quest to find out what new findings this new study presents or add to the existing body of literature. The identified gap could be communicated in the abstract to catch the attention of readers who focus on the abstract first.

Abstract: Please briefly report on the study design, sampling approach and data analysis done in the abstract.

Because this is a quantitative study, include in the results the relevant statistics (mean, SD, correlation coefficient, P-values etc.)

Methods:

1. the study design used was not mentioned

2. the sampling approach used in recruiting the students from the various class or schools was/were not described.

3. A Justification and adequateness of the sample size used is needed.

Results:

The first write-up under the results be captured as data analysis.

Please report on the statistics in the relevant places in the main text (line 350-359 and 374-409).

Reviewer #2: The current study performed a multi-stage analyses to identify the relationships between exam-related stress in students with dyslexia and their educational support, motivation and reading strategies. This is an important study, and the project analyzed previous data to delineate new findings. The method is strong, and appropriate statistical analyses showed the correlations between the various variables.

Major comment: The authors described participants for their studies, but did not show any gender differences in their results. It could be assumed that motivation and stressors in this population is considerably different between boys and girls, but no such results was mentioned.

Minor comments: Typos, unclear sentences, and wrong prepositions in the introduction section makes reading difficult.

E.g.,

1. "In fact, stress presents as one of the key challenges for educational and therapeutic practice focused on the organization of educational support" unclear sentence

2. "...and who, upon reaching higher grades of primary school have to use their often limited literacy skills in producing written assignments and passing written exams as required in the he curriculum." in the curriculum has extra articles.

3. "Tasks connected reading and writing can be seen as challenges, that trigger negative emotions and constitute potentially demanding situations, requiring educational support." Sentence is unclear.

The manuscript, specially the introduction part needs to be checked for typos and better readability.

6. PLOS authors have the option to publish the peer review history of their article (what does this mean?). If published, this will include your full peer review and any attached files.

Reviewer #1: No

Reviewer #2: No

---

## [Author Response · Author response to Decision Letter 0]

30 Jun 2023

Dear Editor, Dear Reviewers,

We are very grateful to have been given the opportunity to revise our manuscript entitled “Exam stress and the metacognitive strategies of reading in students with dyslexia: the role of motivational mechanisms and educational support” for the PLOS ONE journal (Manuscript ID: PONE-D-23-09745). 

We have carefully read the guidelines and used the style templates provided. We hope that the manuscript meets PLOS ONE’s style requirements.

We would like to thank the Reviewers for all valuable comments that helped us to improve our manuscript and contributed to its quality. We have made every effort to respond to each reviewer's suggestion and we hope that we have clarified all your concerns. Below we provide detailed responses to the Reviewers' comments.

Best Regards, 

Authors

Reviewer #1

Reviewer #1: This is an excellent study. Valid methods were used in the identification of dyslexia. The authors clearly stated the gap identified in the literature, the need for this study and its implication on the education of students with dyslexia both in Poland and the world. However, the statement on the justification for this study is not strategically positioned in the manuscript to draw readers attention, arouse their curiosity and sustain their quest to find out what new findings this new study presents or add to the existing body of literature. The identified gap could be communicated in the abstract to catch the attention of readers who focus on the abstract first. Abstract: Please briefly report on the study design, sampling approach and data analysis done in the abstract. Because this is a quantitative study, include in the results the relevant statistics (mean, SD, correlation coefficient, P-values etc.)

Authors’ response: Thank you for your appreciation. The Abstract has been modified according to the reviewer’s comments. We hope that in this form we have better exposed the information important for readers. 

Reviewer #1: Methods: 1. the study design used was not mentioned

Authors’ response: Thank you for this suggestion. The information has been supplemented in accordance with the Reviewer's comment.

Reviewer #1: Methods: 2. the sampling approach used in recruiting the students from the various class or schools was/were not described.

Authors’ response: Thank you for this suggestion. The information has been supplemented in accordance with the Reviewer's comment.

Reviewer #1: Methods: 3. A Justification and adequateness of the sample size used is needed.

Authors’ response: Thank you for this suggestion. The information has been supplemented in accordance with the Reviewer's comment.

Reviewer #1: Results: The first write-up under the results be captured as data analysis.

Authors’ response: Thank you for this suggestion. We have separated the 'Data analysis' section in accordance with the reviewer's comment.

Reviewer #1: Results: Please report on the statistics in the relevant places in the main text (line 350-359 and 374-409).

Authors’ response: Thank you for this suggestion. The information has been supplemented in accordance with the Reviewer's comment.

Reviewer #2

Reviewer #2: Major comment: The authors described participants for their studies, but did not show any gender differences in their results. It could be assumed that motivation and stressors in this population is considerably different between boys and girls, but no such results was mentioned.

Authors’ response: Thank you for this suggestion. Taking gender differences into account can be an important contribution to further in-depth analyzes of the material we have collected. However, in this part of the research project, we wanted to identify the mechanisms of functioning of dyslexic students in a situation of exam stress in relation to the entire study group, regardless of gender. Some identified gender differences will be presented in another manuscript.

Reviewer #2: Minor comments: Typos, unclear sentences, and wrong prepositions in the introduction section makes reading difficult.E.g.,

1. "In fact, stress presents as one of the key challenges for educational and therapeutic practice focused on the organization of educational support" unclear sentence

2. "...and who, upon reaching higher grades of primary school have to use their often limited literacy skills in producing written assignments and passing written exams as required in the he curriculum." in the curriculum has extra articles.

3. "Tasks connected reading and writing can be seen as challenges, that trigger negative emotions and constitute potentially demanding situations, requiring educational support." Sentence is unclear.

The manuscript, specially the introduction part needs to be checked for typos and better readability.

Authors’ response: We read the manuscript carefully and made every effort to eliminate all Typos and unclear sentences.

Reviewer #2: 3. Have the authors made all data underlying the findings in their manuscript fully available? The PLOS Data policy requires authors to make all data underlying the findings described in their manuscript fully available without restriction, with rare exception (please refer to the Data Availability Statement in the manuscript PDF file). The data should be provided as part of the manuscript or its supporting information, or deposited to a public repository. For example, in addition to summary statistics, the data points behind means, medians and variance measures should be available. If there are restrictions on publicly sharing data—e.g. participant privacy or use of data from a third party—those must be specified.

Authors’ response: Data availability information is provided at the end of the manuscript.

---

## [Decision Letter · Decision Letter 1]

9 Aug 2023

PONE-D-23-09745R1Exam stress and the metacognitive strategies of reading in students with dyslexia: the role of motivational mechanisms and educational supportPLOS ONE

Dear Dr. Kuracki,

Thank you for submitting your manuscript to PLOS ONE. After careful consideration, we feel that it has merit but does not fully meet PLOS ONE’s publication criteria as it currently stands. I suggestt that authors carefully reponse to the comments from both reviewers. Therefore, we invite you to submit a revised version of the manuscript that addresses the points raised during the review process.

We look forward to receiving your revised manuscript.

Kind regards,

Yuh-Yuh Li, Ph.D.

Academic Editor

PLOS ONE

Journal Requirements:

Reviewers' comments:

Reviewer's Responses to Questions

**Comments to the Author**

1. If the authors have adequately addressed your comments raised in a previous round of review and you feel that this manuscript is now acceptable for publication, you may indicate that here to bypass the “Comments to the Author” section, enter your conflict of interest statement in the “Confidential to Editor” section, and submit your "Accept" recommendation.

Reviewer #1: (No Response)

Reviewer #3: (No Response)

2. Is the manuscript technically sound, and do the data support the conclusions?

Reviewer #1: Yes

Reviewer #3: Yes

3. Has the statistical analysis been performed appropriately and rigorously? 

Reviewer #1: Yes

Reviewer #3: I Don't Know

4. Have the authors made all data underlying the findings in their manuscript fully available?

Reviewer #1: Yes

Reviewer #3: Yes

5. Is the manuscript presented in an intelligible fashion and written in standard English?

Reviewer #1: Yes

Reviewer #3: No

6. Review Comments to the Author

Reviewer #1: Based on the information provided by the authors on page 12, Line 276-278, it is still not clear how the authors arrived at the sample size 640. Did they use the Taro Yamane formular, Krejcie & Mogan formular or the Cochrane formular. Again with a 95% confidence interval, the margin of error or the acceptable sampling error is expected to be 5% but they quoted 4%. When you tabulate the values provided by the authors into the Yamane and Krejcie & Mogan formular, you still do not arrive at their estimated sample size (64). if these calculations were not factored into the determination of the sample size prior to the study, i suggest they discuss that as a methodological limitation in their study.

Reviewer #3: The authors have done thorough literature review on the subject matter. Below are my observations and suggestions after review

1. Line 33. If you are referring to your paper, I suggest “the study showed” to “studies” to avoid any comprehension issues to the readers.

2. Both British and American spellings are employed; perhaps adhering to one standard would be optimal.

3. Line 131, no citation for the findings reported

4. Full stop after the in text citation 52 and 77

5. Line 135 going “Yet, it appears that students with dyslexia have comparable insight into their own difficulties and are able to adjust their reading comprehension expectations, much like students without any specific reading difficulties. Both groups show similar sensitivity to metacognitive reading experiences.” Support the claim with evidence or citation

6. Line 167 “Motivational” seems not to fit in the sentence perhaps “motivation” could be used.

7. 179 to 182 “Research shows that…” which research? citation

8. Line 132 to 142 and Line 179 to 190 are just repetition of same ideas and words.

9. Why was other aspects of validity analysis not performed or reported to fully validate the scale but only reliability scores. Those analysis could be reported as supplementary materials.

10. Whether analysis on crucial regression assumptions was carried out and met or not was not reported or mentioned to conclude whether the analysis outcome was reliable or not.

11. The work could benefit from additional professional editing.

7. PLOS authors have the option to publish the peer review history of their article (what does this mean?). If published, this will include your full peer review and any attached files.

Reviewer #1: No

Reviewer #3: No

---

## [Author Response · Author response to Decision Letter 1]

17 Oct 2023

Dear Editor, Dear Reviewers,

We are very grateful to have been given the opportunity to revise our manuscript entitled “Exam stress and the metacognitive strategies of reading in students with dyslexia: the role of motivational mechanisms and educational support” for the PLOS ONE journal (Manuscript ID: PONE-D-23-09745). 

We have carefully read the guidelines and used the style templates provided. We hope that the manuscript meets PLOS ONE’s style requirements.

We would like to thank the Reviewers for all valuable comments that helped us to improve our manuscript and contributed to its quality. We have made every effort to respond to each reviewer's suggestion and we hope that we have clarified all your concerns. Below we provide detailed responses to the Reviewers' comments.

Best Regards, 

Authors

Reviewer #1: Based on the information provided by the authors on page 12, Line 276-278, it is still not clear how the authors arrived at the sample size 640. Did they use the Taro Yamane formular, Krejcie & Mogan formular or the Cochrane formular. Again with a 95% confidence interval, the margin of error or the acceptable sampling error is expected to be 5% but they quoted 4%. When you tabulate the values provided by the authors into the Yamane and Krejcie & Mogan formular, you still do not arrive at their estimated sample size (64). if these calculations were not factored into the determination of the sample size prior to the study, i suggest they discuss that as a methodological limitation in their study.

Authors’ response: Thank you for this suggestion. There was a typo in the text. Instead of 4% it should be 5%, which has been corrected. There was a typo in the text. Instead of 4% it should be 5%, which has been corrected. Using the Taro Yamane formula (for N = 5,000,000 Polish students with dyslexia and 95% confidence interval), the minimum size of the study sample for the population of students with dyslexia in Poland should be 400 people. This condition has therefore been met. Due to the fact that it was possible to collect a larger number of respondents who met the criteria, the authors decided to perform analyzes on data from a larger sample.

Reviewer #3: The authors have done thorough literature review on the subject matter. Below are my observations and suggestions after review

1. Line 33. If you are referring to your paper, I suggest “the study showed” to “studies” to avoid any comprehension issues to the readers.

Authors’ response: Thank you for this suggestion. The change suggested by the Reviewer was made.

2. Both British and American spellings are employed; perhaps adhering to one standard would be optimal.

Authors’ response: Thank you for this suggestion. The change suggested by the Reviewer was made.

3. Line 131, no citation for the findings reported

Authors’ response: Thank you for this suggestion. The information has been supplemented in accordance with the Reviewer's comment.

4. Full stop after the in text citation 52 and 77

Authors’ response: Thank you for this suggestion. The change suggested by the Reviewer was made.

5. Line 135 going “Yet, it appears that students with dyslexia have comparable insight into their own difficulties and are able to adjust their reading comprehension expectations, much like students without any specific reading difficulties. Both groups show similar sensitivity to metacognitive reading experiences.” Support the claim with evidence or citation

Authors’ response: Thank you for this suggestion. The information has been supplemented in accordance with the Reviewer's comment.

6. Line 167 “Motivational” seems not to fit in the sentence perhaps “motivation” could be used.

Authors’ response: Thank you for this suggestion. The change suggested by the Reviewer was made.

7. 179 to 182 “Research shows that…” which research? citation

Authors’ response: Thank you for this suggestion. The information has been supplemented in accordance with the Reviewer's comment.

8. Line 132 to 142 and Line 179 to 190 are just repetition of same ideas and words.

Authors’ response: Thank you for this valuable suggestion. We removed the repetitive part from the text.

9. Why was other aspects of validity analysis not performed or reported to fully validate the scale but only reliability scores. Those analysis could be reported as supplementary materials.

In the manuscript, we limited ourselves to providing only the reliability indicators of the research tools used for the studied sample, which is consistent with the practice of preparing this type of texts, the main goal of which is not to describe the adaptation of the tools. One of the tools - The Educational Support Questionnaire - has a full description of its validation in the Polish group in a separate article, included in the bibliographic list [67]. In the case of the next tool - Metacognitive Awareness of Reading Strategies Inventory (MARSI) - the adaptation was made for the needs of other studies conducted in a group of Polish children, and the results were published in Dłużniewska A. (2021). Rozumienie tekstów literackich przez uczniów z uszkodzeniami słuchu. Kraków: Oficyna Wydawnicza Impuls. A full description of the validation, including content, theoretical and predictive validity of the other two questionnaires (on a larger research sample) was prepared for the needs of separate articles that are currently in the publishing process, which makes it impossible to include them as supplementary material in this text.

10. Whether analysis on crucial regression assumptions was carried out and met or not was not reported or mentioned to conclude whether the analysis outcome was reliable or not.

Authors’ response: Thank you for this suggestion. The information has been supplemented in accordance with the Reviewer's comment.

11. The work could benefit from additional professional editing.

Authors’ response: We have made every effort to prepare a revised manuscript in accordance with the publishing house's editorial requirements. Attached we also send a certificate of proofreading performed by a native speaker.

---

## [Editor Report · Decision Letter 2]

30 Oct 2023

Exam stress and the metacognitive strategies of reading in students with dyslexia: the role of motivational mechanisms and educational support

PONE-D-23-09745R2

Dear Dr. Kuracki,

We’re pleased to inform you that your manuscript has been judged scientifically suitable for publication and will be formally accepted for publication once it meets all outstanding technical requirements.

Kind regards,

Yuh-Yuh Li, Ph.D.

Academic Editor

PLOS ONE
---

## [Editor Report · Acceptance letter]

3 Nov 2023

PONE-D-23-09745R2 

Exam stress and the metacognitive strategies of reading in students with dyslexia: the role of motivational mechanisms and educational support 

Dear Dr. Kuracki:

I'm pleased to inform you that your manuscript has been deemed suitable for publication in PLOS ONE. Congratulations! Your manuscript is now with our production department. 

Kind regards, 

on behalf of

Dr. Yuh-Yuh Li 

Academic Editor

PLOS ONE